# AutoMTL: A Programming Framework for Automating Efficient Multi-Task Learning

**Lijun Zhang**
College of Information & Computer Sciences
University of Massachusetts Amherst
Amherst, MA, 01003
`lijunzhang@cs.umass.edu`

**Xiao Liu**
College of Information & Computer Sciences
University of Massachusetts Amherst
Amherst, MA, 01003
`xiaoliu1990@cs.umass.edu`

**Hui Guan**
College of Information & Computer Sciences
University of Massachusetts Amherst
Amherst, MA, 01003
`huiguan@cs.umass.edu`

## Abstract

Multi-task learning (MTL) jointly learns a set of tasks by sharing parameters among tasks. It is a promising approach for reducing storage costs while improving task accuracy for many computer vision tasks. The effective adoption of MTL faces two main challenges. The first challenge is to determine what parameters to share across tasks to optimize for both memory efficiency and task accuracy. The second challenge is to automatically apply MTL algorithms to an arbitrary CNN backbone without requiring time-consuming manual re-implementation and significant domain expertise. This paper addresses the challenges by developing the first programming framework AutoMTL that automates efficient MTL model development for vision tasks. AutoMTL takes as inputs an arbitrary backbone convolutional neural network (CNN) and a set of tasks to learn, and automatically produces a multi-task model that achieves high accuracy and small memory footprint simultaneously. Experiments on three popular MTL benchmarks (CityScapes, NYUv2, Tiny-Taskonomy) demonstrate the effectiveness of AutoMTL over state-of-the-art approaches as well as the generalizability of AutoMTL across CNNs. AutoMTL is open-sourced and available at `https://github.com/zhanglijun95/AutoMTL`.

## 1 Introduction

AI-powered applications increasingly adopt Convolutional Neural Networks (CNNs) for solving many vision-related tasks (e.g., semantic segmentation, object detection), leading to more than one CNNs running on resource-constrained devices. Supporting many models simultaneously on a device is challenging due to the linearly increased computation, energy, and storage costs. An effective approach to address the problem is multi-task learning (MTL) where a set of tasks are learned jointly to allow some parameter sharing among tasks. MTL creates *multi-task models* based on CNN architectures called *backbone models*, and has shown significantly reduced inference costs and improved generalization performance in many computer vision applications [24, 37].

The effective adoption of MTL faces two main challenges. The first challenge is the resource-efficient architecture design–that is, to determine what parameters of a backbone model to share across tasks to optimize for both resource efficiency and task accuracy. Many prior works [20, 22, 13, 24, 36] rely on manually-designed MTL model architectures which share several initial layers and then branch

36th Conference on Neural Information Processing Systems (NeurIPS 2022).

out at an ad hoc point for all tasks. They often result in unsatisfactory solutions due to the enormous architecture search space. Several recent efforts [43, 1, 19] shift towards learning to share parameters across tasks. They embed policy-learning components into a backbone CNN and train the policy to determine which blocks in the network should be shared across which task or where to branch out for different tasks. Their architecture search spaces lack the flexibility to dynamically adjust model capacity based on given tasks, leading to sub-optimal solutions as the number of tasks grows.

The second major challenge is the automation. Manual architecture design calls for significant domain expertise when tweaking neural network architectures for every possible combination of learning tasks. Although neural architecture search (NAS)-based approaches automate the model design to some extent, the implementation of these works is deeply coupled with a specific backbone model. Some of them [1, 43, 4, 19] could theoretically support broader types of CNNs. They, however, require significant manual efforts and expertise to re-implement the proposed algorithms whenever the backbone changes. Our user study (Section 4.2) suggests that it takes machine learning practitioners with proficient PyTorch skills 20 to 40 hours to re-implement Adashare [43], a state-of-the-art NAS-based MTL approach, on a MobileNet backbone. The learning curve is expected to be much longer and more difficult for general programmers with less ML expertise. The lack of automation prohibits the effective adoption of MTL in practice.

In this paper, we address the two challenges by developing AutoMTL, the first programming framework that automates resource-efficient MTL model development for vision tasks. AutoMTL takes as inputs an arbitrary backbone CNN and a set of tasks to learn, and then produces a multi-task model that achieves high task accuracy and small memory footprint (measured by the number of parameters). A backbone CNN essentially specifies a computation graph where each node is an operator. The key insight is to treat each operator as a basic unit for sharing. Each task can select which operators to use to determine the sharing patterns with other tasks. The operator-level sharing granularity not only enables the automatic support of arbitrary CNN backbone architectures, but also leads to a stretchable architecture search space that contains multi-task models with a wide range of model capacity. To our best knowledge, we are the first work considering parameter sharing in MTL at the operator level.

AutoMTL features a source-to-source compiler that automatically transforms a user-provided backbone CNN to a supermodel that encodes the multi-task architecture search space in the operator-level granularity. It also offers a set of PyTorch-based APIs to allow users to flexibly specify the input backbone model. AutoMTL then performs gradient-based architecture searches on the supermodel to identify the optimal sharing patterns among tasks. Our experiments on several MTL benchmarks with a different number of tasks show that AutoMTL could produce efficient multi-task models with smaller memory footprint and higher task accuracy compared to state-of-the-art methods. AutoMTL also automatically supports arbitrary CNN backbones without any re-implementation efforts and improves the accessibility of MTL to general programmers.

The main contributions of our work are as follows:

- We propose a *Multi-Task Supermodel Compiler* (MTS-Compiler), which transforms a user-provided backbone CNN into a multi-task supermodel that encodes the architecture search space. The compiler decouples architecture search with the backbone CNN, removing the manual efforts in re-implementing MTL on new backbone models.
- We propose a *Stretchable Architecture Search Space* that offers flexibility in deriving multi-task models with a wide range of model capacity based on task difficulties and interference. We further propose a novel data structure called *Virtual Computation Node* to encode the search space and enable compiler-based multi-task supermodel transformation.
- Built on top of the supermodel, we adopt the Gumbel-Softmax approximation and standard back-propagation to jointly optimize sharing policies and network weights. Under this context, we propose *a policy regularization term* on the sharing policy to promote parameter sharing for high memory efficiency.
- We implement the AutoMTL system that seamlessly integrates the MTS-Compiler, a set of PyTorch-based APIs, the architecture search algorithm, and a training pipeline. AutoMTL provides an easy-to-use solution for resource-efficient multi-task model development.
- Experiments on three popular MTL benchmarks (CityScapes [10], NYUv2 [41], Tiny-Taskonomy [50]) using three common CNNs (Deeplab-ResNet34 [7], MobileNetV2 [39], MNasNet [45]) demonstrate that the multi-task model produced by AutoMTL outperforms state-of-the-art approaches in terms of model size and task accuracy.

## 2   Related Work

Multi-task learning (MTL) uses hard or soft parameter sharing [37, 5, 2, 47, 52]. In hard parameter sharing, a set of parameters in the backbone model are shared among tasks. In soft parameter sharing [34, 38, 18], each task has its own set of parameters. Task information is shared by enforcing the weights of the model for each task to be similar. In this paper, we focus on hard parameter sharing as it produces memory-efficient multi-task models.

**Manual Design and Task Grouping.** One of the widely-used hard parameter sharing strategies is proposed by Caruana [6, 5], which shares the bottom layers of a model across tasks. Following this paradigm, early works [32, 35, 44, 9, 25, 26] rely on domain expertise to decide which layers should be shared across tasks and which ones should be task-specific. Due to the enormous architecture search space, such approaches are difficult to find an optimal solution. In recent years, several methods attempt to integrate task relationships or similarities to facilitate multi-task model design. What-to-Share [46] measures the task affinity by analyzing the representation similarity between independent models, then recommending the architecture with the minimum total task dissimilarity. Some other works [42, 16] focus on identifying the best task groupings in terms of task performance under the memory budget, whose architectures share the feature extractors within each group.

**NAS-based Approach.** Recent works attempt to *learn* the sharing patterns across tasks. Deep Elastic Network (DEN) [1] and Stochastic Filter Groups (SFGs) [3] determine whether each filter in convolutions should be shared via reinforcement learning or variational inference respectively. AdaShare [43] learns task-specific policies that select which network blocks to execute for a given task. However, these works cannot dynamically adapt multi-task model capacity based on given tasks. BMTAS [4] and Learn to Branch [19] focus on constructing tree-like structures for multi-task models via differentiable neural architecture search. Some other works [17, 48] explore feature fusion opportunities across tasks. It is worth mentioning that existing studies focus on search algorithms but ignore the ease of programming and extensibility. Their implementation is highly coupled with specific backbone models. It is time-consuming for general programmers to re-implement the algorithms once the backbone models change, hindering their adoption in broader applications.

**MTL Optimization.** Significant efforts have been invested to improve multi-task optimization strategies, an orthogonal direction to architecture design. There are two major branches to solve such a multi-objective optimization problem [40, 28]. Some works study a single surrogate loss consisting of linear combination of task losses, in which the suitable task weights are derived from different criteria, such as task uncertainty[23], task loss magnitudes [31], dynamic task relationships [30]. Other works focus on directly modifying task gradients during the multi-task model training [8, 49, 29]. Note that, on top of our AutoMTL framework, users are free to use existing optimization methods to further improve the task performance of multi-task models.

## 3   AutoMTL

AutoMTL allows users to provide an arbitrary backbone CNN and a set of vision tasks, and then automatically generate a multi-task model with high accuracy and low memory footprint by sharing parameters among tasks. Figure 1 illustrates the workflow of AutoMTL. Given a backbone model (Figure 1(a)), a user can specify the model using either the AutoMTL APIs or in the *prototxt* format (Figure 1(b)). The model specification will be parsed by the *MTS-compiler* to generate a multi-task supermodel that encodes the entire search space (Figure 1(c)). AutoMTL then identifies the optimal multi-task model architecture (Figure 1(d)) from the supermodel using gradient-based architecture search algorithms implemented in the *Architecture Search* component. AutoMTL supports the model specification in the format of *prototxt* as it is general enough to support various CNN architectures and also simple for our compiler to analyze. *prototxt* files are serialized using Google's Protocol Buffers serialization library. AutoMTL API is currently implemented on top of PyTorch. We next elaborate on the two major components of this framework, *MTS-Compiler* and *Architecture Search*.

### 3.1   Multi-Task Supermodel Compiler

The Multi-Task Supermodel Compiler (MTS-Compiler) transforms the input backbone model into a *multi-task supermodel* that encodes the architecture search space. The challenge is how to design the architecture search space and the multi-task supermodel so that (1) the search space allows the

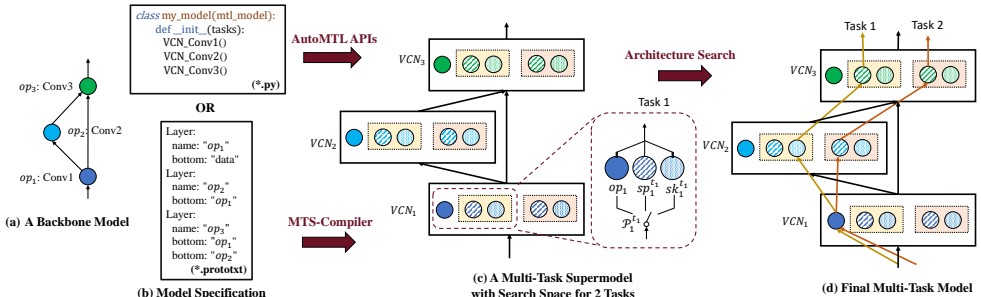

Figure 1: Illustrations of (a) an input backbone model, (b) two types of model specifications, (c) the multi-task supermodel with search space produced by our AutoMTL APIs and the proposed MTS-Compiler, and (d) the final multi-task model found by AutoMTL. $VCN_1 \sim VCN_3$ represent the proposed data structure Virtual Computation Node (VCN). In each VCN, $op$ is the original operator in the backbone model, while $sp^{t_i}$ and $sk^{t_i}$ are the task-specific copy of $op$ and the skip connection for task $t_i$ respectively. $\mathcal{P}^{t_i}$ is a variable (a.k.a policy) that determines which operator will be executed for task $t_i$.

multi-task model capacity to be *flexibly adjusted* based on the set of tasks to avoid task interference and (2) the transformation can be *fully automated* and support an arbitrary CNN backbone.

To address the challenge, we propose a *Stretchable Architecture Search Space* that contains multi-task models with a wide range of model capacities by treating each operator in the backbone model as the basic unit for sharing. The compiler duplicates each operator in the backbone model so that each task can determine whether it wants to share parameters with other tasks by selecting which operator to use. We further design a novel data structure called *Virtual Computation Node* to embed the search space and enable compiler-based automatic transformation of an arbitrary CNN to a multi-task supermodel.

**Stretchable Architecture Search Space.** As shown in Figure 1(c), for each operator in the backbone model, each task can choose from one of the three options to indicate whether it wants to share the operator with other tasks: (1) the *backbone operator*, (2) a *task-specific copy* of the backbone operator, and (3) a *skip connection*. Skip connection is an identify function if the input and output dimension match or a down-sample function otherwise. The motivation for a skip connection option is that a task can skip the operator to improve inference efficiency.

Formally, assume a set of $N$ tasks $\mathbf{T} = \{t_1, t_2, ..., t_N\}$ defined over a dataset. For the $i$-th task $t_i$ and the $l$-th backbone operator $op_l$, the task may select the backbone operator itself, implying that it can share the parameters in this operator with other tasks. Otherwise, it can select the task-specific copy $sp_l^{t_i}$, or the skip connection $sk_l^{t_i}$, as shown in Figure 1(c). Given a set of $N$ tasks and a backbone model with $L$ operators, the size of our search space is $3^{N \times L}$. Suppose the backbone capacity is $C$ (measured by the number of parameters), the capacity of a multi-task model in our search space would be in the range of $(0, C \times N]$, where $C \times N$ indicates all tasks choose to use their own operators (i.e., independent models) while 0 represents all the skip connections are selected.

The proposed search space has the following three major benefits compared to existing NAS-based MTL methods. First, compared to AdaShare [43] and DEN [1] which attempt to pack multiple tasks into a single CNN backbone, it can extend the representation power of the backbone model if needed by preserving more task-specific operators. This capability effectively avoids performance degradation caused by task interference as the number of tasks increases (See Section 4.2). Second, compared to a more general search space [48] that is defined without requiring a user-provided backbone model, it still provides users a certain degree of control over the size of the searched multi-task model – one can specify a smaller backbone model if the computation resource is limited. Last but not least, in terms of search efficiency, although our search space is larger than AdaShare, we still have a comparable low search cost by adopting gradient-based search algorithms. When comparing to a general search space in FBNetV5 [48], our search space can be explored $13 \sim 133X$ faster (Detailed in Section 4.2).

**Multi-Task Supermodel.** A suitable multi-task supermodel abstraction is necessary to allow *automatic* transformation of an arbitrary CNN backbone to a multi-task supermodel. Our idea is to represent the multi-task supermodel as a computation graph whose topology remains the same as that of the backbone model but nodes are replaced with Virtual Computation Nodes (VCNs). The

MTS-Compiler first parses the input backbone into a list of operators and then iterates the operators to initialize the corresponding VCNs.

Specifically, for each operator in the given backbone model, the corresponding VCN in the multi-task supermodel contains: (1) a list of parent VCN nodes, recording where inputs come from; (2) the backbone operator; (3) task-specific copies of the backbone operator, one for each task; (4) skip connections, one for each task; (5) policy variables, one for each task determining which operator to execute for each task. (1) and (2) encode the computation graph of the backbone model. (2), (3), and (4) together encode the architecture search space. (5) determines the sharing patterns across tasks. Figure 1(c) illustrates a multi-task supermodel. Details about the policy variable will be presented in Section 3.2.

## 3.2 Architecture Search

The architecture search component aims at efficiently exploring the search space encoded in the multi-task supermodel. We adopt the *differentiable policy approximation* to enable joint training of sharing policy and the supermodel. Under this context, we propose a *policy regularization* mechanism to promote parameter sharing for memory efficiency.

**Policy Approximation.** We introduce a trainable policy variable $\mathcal{P}_l^{t_i}$ to determine which operator to use for the $i$-th task $t_i$ and the $l$-th VCN in the multi-task supermodel. $\mathcal{P}_l^{t_i}$ is zero if the backbone operator $op_l$ is used for the task, one if the task-specific copy $sp_l^{t_i}$ is adopted, and two if the skip connection $sk_l^{t_i}$ is selected. Architecture search is to find the optimal sharing policy $\mathbf{P} = \{\mathcal{P}_l^{t_i} | l \leq L, i \leq N\}$, that yields the best overall performance over the set of $N$ tasks $\mathbf{T}$ given a multi-task supermodel with $L$ VCNs. As the number of potential configurations for $\mathbf{P}$ is $3^{N \times L}$ (i.e., the size of the search space) which grows exponentially with the number of operators and tasks, it is not practical to manually find such a $\mathbf{P}$ to get the optimal sharing pattern. Therefore, we adopt a gradient-based architecture search algorithm that optimizes the sharing policy $\mathbf{P}$ and the multi-task model parameters jointly via standard back-propagation. Gradient-based searches usually allow faster architecture search compared with traditional reinforcement learning [1] or evolutionary algorithm-based approaches [27].

Because the policy variable $\mathcal{P} \in \mathbf{P}$ is discrete and thus non-differentiable, we apply Gumbel-Softmax Approximation [21] and derives a soft differentiable policy:

$$\mathcal{P}'(k) = \frac{\exp((G_k + \log(\pi_k))/\tau)}{\sum_{k \in \{0,1,2\}} \exp((G_k + \log(\pi_k))/\tau)}, \tag{1}$$

where $k \in \{0, 1, 2\}$ represents the three operator options, $G_k \sim Gumbel(0, 1)$.

After learning the distribution $\pi$, the discrete task-specific policy $\mathcal{P}$ is sampled from the learned $\pi$ to decide which operator to execute in each VCN for each task and the multi-task architecture can be constructed accordingly. Figure 1(d) illustrates a multi-task model given a sharing policy.

**Policy Regularization.** We propose a policy regularization term $\mathcal{L}_{reg}$ to encourage sharing operators across tasks to reduce the memory overhead. Specifically, for the soft policy $\mathbf{P}' = \{\mathcal{P}_l'^{t_i} | l \leq L, i \leq N\}$, we minimize the sum of the SoftPlus [12] of $\mathcal{P}_l'^{t_i}(1) - \mathcal{P}_l'^{t_i}(0)$ and $\mathcal{P}_l'^{t_i}(2) - \mathcal{P}_l'^{t_i}(0)$ for each task in each VCN, where $\mathcal{P}_l'^{t_i}(0), \mathcal{P}_l'^{t_i}(1), \mathcal{P}_l'^{t_i}(2)$ are the probability of selecting the shared operator, the task-specific copy, and the skip connection for the $i$-th task in the $l$-th VCN respectively. To further reduce the computation cost, the regularization term is weighted for different operators to promote the parameter sharing of bottom layers. More formally, we define $\mathcal{L}_{reg}$ as,

$$\mathcal{L}_{reg} = \sum_{i \leq N} \sum_{l \leq L} \frac{L-l}{L} \left\{ \ln(1 + \exp^{\mathcal{P}_l'^{t_i}(1) - \mathcal{P}_l'^{t_i}(0)}) + \ln(1 + \exp^{\mathcal{P}_l'^{t_i}(2) - \mathcal{P}_l'^{t_i}(0)}) \right\}, \tag{2}$$

where $\ln(1 + \exp^x)$ is the SoftPlus function. $l$ is the depth of the current VCN and $L$ is the maximum depth. Finally, the overall loss $\mathcal{L}$ is defined as,

$$\mathcal{L} = \sum_i \lambda_i \mathcal{L}_i + \lambda_{reg} \mathcal{L}_{reg}, \tag{3}$$

where $\mathcal{L}_i$ represents the task-specific loss, $\lambda_i$ is a hyperparameter controlling how much each task contributes to the overall loss, and $\lambda_{reg}$ is a hyper-parameter balancing task-specific losses and $\mathcal{L}_{reg}$.

**Training Pipelines.** AutoMTL implements a three-stage training pipeline to generate a well-trained multi-task model. The first stage *pre-train* aims at obtaining a good initialization for the multi-task supermodel by pre-training on tasks jointly [51]. During pre-training, for each task, the output of each VCN is the average of the three operator options (i.e, the backbone operator, the task-specific copy, and the skip connection) so that all the parameters could get warmed up together. The second stage *policy-train* jointly optimizes the sharing policy and the model parameters. The model parameters and the policy distribution parameters are trained alternately to stabilize the training process. After the policy distribution parameters converge, AutoMTL samples a sharing policy from the distribution to generate a multi-task model. The last stage *post-train* trains the identified multi-task model until it converges. The model parameters are trained from scratch while the sharing policy is fixed.

## 4 Experiments

We conduct a set of experiments to examine the superiority of AutoMTL compared to several state-of-the-art approaches in terms of task accuracy, model size, and inference time.

### 4.1 Experiment Settings

**Datasets and Tasks.** Our experiments use three popular datasets in multi-task learning (MTL), **CityScapes** [10], **NYUv2** [41], and **Tiny-Taskonomy** [50]. CityScapes contains street-view images and two tasks, semantic segmentation and depth estimation. The NYUv2 dataset consists of RGB-D indoor scene images and three tasks, 13-class semantic segmentation defined in [11], depth estimation whose ground truth is recorded by depth cameras from Microsoft Kinect, and surface normal prediction with labels provided in [14]. Tiny-Taskonomy contains indoor images and its five representative tasks are semantic segmentation, surface normal prediction, depth estimation, keypoint detection, and edge detection. All the data splits follow the experimental settings in AdaShare [43].

**Loss Functions and Evaluation Metrics.** Semantic segmentation uses a pixel-wise cross-entropy loss for each predicted class label. Surface normal prediction uses the inverse of cosine similarity between the normalized prediction and ground truth. All other tasks use the L1 loss. Semantic segmentation is evaluated using mean Intersection over Union and Pixel Accuracy (mIoU and Pixel Acc, the higher the better) in both CityScapes and NYUv2. Surface normal prediction is evaluated using mean and median angle distances between the prediction and the ground truth (the lower the better), and the percentage of pixels whose prediction is within the angles of $11.25°$, $22.5°$ and $30°$ to the ground truth as [14] (the higher the better). Depth estimation uses the absolute and relative errors between the prediction and the ground truth are computed (the lower the better). In addition, the percentage of pixels whose prediction is within the thresholds of $1.25, 1.25^2, 1.25^3$ to the ground truth, i.e. $\delta = \max\{\frac{p_{pred}}{p_{gt}}, \frac{p_{gt}}{p_{pred}}\} < thr$, is used following [15] (the higher the better). Tiny-Taskonomy is evaluated using the task-specific loss of each task directly, as in [43].

Because evaluation metrics from different tasks have different scales, we also use a single **relative performance** metric [33] with respect to the single-task baseline to compare different approaches. The relative performance $\Delta t_i$ of a method $A$ on task $t_i$ is computed as follows,

$$\Delta t_i = \frac{1}{|M|} \sum_{j=0} (-1)^{s_j} (M_{A,j} - M_{STL,j})/M_{STL,j} \times 100\%,$$

where $s_j$ is 1 if the metric $M_j$ is the lower the better and 0 otherwise. $M_{A,j}$ and $M_{STL,j}$ are the values of the metric $M_j$ for the method $A$ and the **Single-Task** baseline respectively. Besides, the overall performance is the average of the above relative values over all tasks, namely $\Delta t = \frac{1}{N} \sum_{i=1} \Delta t_i$, where $N$ is the number of tasks. The model size is evaluated using the number of model parameters.

**Baselines for Comparison.** We compare with following baselines: the **Single-Task** baseline where each task has its own model and is trained independently, the vanilla **Multi-Task** baseline [5] where all tasks share the backbone model but have separate prediction heads, popular MTL methods (e.g., **Cross-Stitch** [34], **Sluice** [38], **NDDR-CNN** [18], **MTAN** [31]), and state-of-the-art NAS-based MTL methods (e.g. **DEN** [1], **AdaShare** [43], and **Learn to Branch**[1] [19]). We use the same backbone

---

[1]We implemented its tree-structured multi-task model for Taskonomy based on the architecture reported in the paper by ourselves since there is no public code.

Table 1: Quantitative Results on CityScapes. (Abs. Prf. & Rel. Prf.)

| Model | # Params ↓ | | Semantic Seg. | | | Depth Estimation | | | | | | Δt ↑ |
|---|---|---|---|---|---|---|---|---|---|---|---|---|
| | Abs. (M) | Rel. (%) | mIoU ↑ | Pixel Acc. ↑ | Δt₁ ↑ | Error ↓ | | δ, within ↑ | | | Δt₂ ↑ | |
| | | | | | | Abs. | Rel. | 1.25 | 1.25² | 1.25³ | | |
| Single-Task | 42.569 | - | 36.5 | 73.8 | - | 0.026 | 0.38 | 57.5 | 76.9 | 87.0 | - | - |
| Multi-Task | 21.285 | -50.0 | 42.7 | 68.1 | +4.6 | 0.026 | 0.39 | 58.8 | 80.5 | 89.9 | +1.5 | +3.1 |
| Cross-Stitch | 42.569 | +0.0 | 40.3 | 74.3 | +5.5 | **0.017** | 0.34 | 70.0 | 86.3 | 93.1 | **+17.2** | +11.4 |
| Sluice | 42.569 | +0.0 | 39.8 | 74.2 | +4.8 | 0.018 | 0.35 | 68.9 | 85.8 | 92.8 | +15.3 | +10.1 |
| NDDR-CNN | 44.059 | +3.5 | 41.5 | 74.2 | +7.1 | 0.018 | 0.35 | 69.9 | 86.3 | 93.0 | +15.9 | +11.5 |
| MTAN | 51.296 | +20.5 | 40.8 | 74.3 | +6.2 | **0.017** | 0.36 | 71.0 | 86.3 | 92.8 | +16.4 | +11.3 |
| DEN | 23.838 | -44.0 | 38.0 | 74.2 | +2.3 | 0.018 | 0.41 | 68.2 | 84.5 | 91.6 | +11.3 | +6.8 |
| AdaShare | 21.285 | -50.0 | 40.6 | 74.7 | +6.2 | 0.018 | 0.37 | **71.4** | **86.8** | 93.1 | +15.5 | +10.9 |
| AutoMTL | 28.819 | -32.3 | **42.8** | **74.8** | **+9.3** | 0.018 | **0.33** | 70.0 | 86.6 | **93.4** | +17.1 | **+13.2** |

Table 2: Results on NYUv2. (Rel. Prf.)

| Model | # Params (%) ↓ | Δt₁ ↑ | Δt₂ ↑ | Δt₃ ↑ | Δt ↑ |
|---|---|---|---|---|---|
| Multi-Task | -66.7 | -11.4 | +2.0 | +4.3 | -1.7 |
| Cross-Stitch | +0.0 | -2.6 | +8.7 | +3.9 | +3.3 |
| Sluice | +0.0 | -6.2 | +7.1 | +3.3 | +1.4 |
| NDDR-CNN | +5.0 | -12.9 | +7.0 | -4.4 | -3.5 |
| MTAN | +3.7 | -1.8 | **+11.5** | +2.9 | +4.2 |
| DEN | -62.7 | -7.7 | +5.6 | -38.9 | -13.7 |
| AdaShare | -66.7 | -4.3 | +9.3 | +6.2 | +3.8 |
| AutoMTL | -45.1 | **+0.2** | +8.0 | **+7.8** | **+5.3** |

Table 3: Results on Taskonomy. (Rel. Prf.)

| Models | # Params (%) ↓ | Δt₁ ↑ | Δt₂ ↑ | Δt₃ ↑ | Δt₄ ↑ | Δt₅ ↑ | Δt ↑ |
|---|---|---|---|---|---|---|---|
| Multi-Task | -80.0 | -3.7 | -1.4 | -4.5 | +0.0 | +4.2 | -1.1 |
| Cross-Stitch | +0.0 | +0.9 | -3.5 | **+0.0** | -1.0 | -2.4 | -1.2 |
| Sluice | +0.0 | -3.7 | -1.5 | -9.1 | +0.5 | +2.4 | -2.3 |
| NDDR | +8.2 | -4.2 | -0.9 | -4.5 | +0.5 | +4.2 | -1.0 |
| MTAN | -9.8 | -8.0 | -2.5 | -4.5 | +0.0 | +2.8 | -2.4 |
| DEN | -77.6 | -28.2 | -2.6 | -22.7 | +2.5 | +4.2 | -9.3 |
| AdaShare | -80.0 | +2.3 | -0.6 | -4.5 | **+3.0** | +5.7 | +1.2 |
| Learn to B. | -71.2 | **+9.4** | +5.3 | -4.5 | -2.5 | -2.4 | +1.1 |
| AutoMTL | -50.1 | +3.0 | **+8.2** | **+0.0** | **+3.0** | **+7.1** | **+4.3** |

$t_1$: Semantic Seg., $t_2$: Surface Normal, $t_3$: Depth Est., $t_4$: Keypoint Det., $t_5$: Edge Det..

model in all baselines and in our approach for fair comparisons. We use Deeplab-ResNet-34 as the backbone model and the Atrous Spatial Pyramid Pooling (ASPP) architecture as the task-specific head [7]. Both of them are popular architectures for pixel-wise prediction tasks. We also evaluate the effectiveness of AutoMTL on MobileNetV2 [39], and MNasNet [45].

### 4.2 Results

**Performance Comparison.** Table 1∼3 report the task performance on each dataset respectively. For CityScapes, both the absolute and the relative performance of all metrics are reported (see Table 1). Due to the limited space, only the relative performance is reported for NYUv2 and Tiny-Taskonomy (see Table 2 and 3).

According to Table 1, AutoMTL outperforms all the baselines on 4 metrics (bold) and is the second-best on 2 metrics (underlined) in terms of task performance. With 17.7% increase in the number of model parameters, the task performance of AutoMTL is far better than the vanilla Multi-Task baseline. Compared to the soft-parameter sharing methods, Cross-Stitch, Sluice, and NDDR-CNN, which are unable to reduce the memory overhead, AutoMTL could achieve higher task performance with fewer parameters. When comparing with DEN and AdaShare, the most competitive approaches in MTL, AutoMTL is better in terms of the task performance but with more parameters (11.7%/17.7%). This is because, unlike DEN and AdaShare which pack tasks into the given backbone model, our search space allows each task to select more task-specific operators to increase the capability of the backbone model. It turns out that a small amount of increase in model parameters could translate to a significant gain in task performance.

The superiority of AutoMTL can be observed more clearly in Tables 2∼3 when more tasks are jointly trained together. AutoMTL outperforms most of the baselines in both task performance and model size. When compared with state-of-the-art NAS-based MTL methods, DEN, AdaShare, and Learn to Branch, AutoMTL could achieve a substantial increase in task accuracy with only a few more parameters. Although DEN and AdaShare need fewer model parameters, the representation power of their multi-task models is limited by the backbone model due to their non-stretchable search space, making it essential for users to select a suitable backbone model with sufficient capacity for multiple tasks. This problem goes worse when the number of tasks increases. As shown in Table 3, DEN

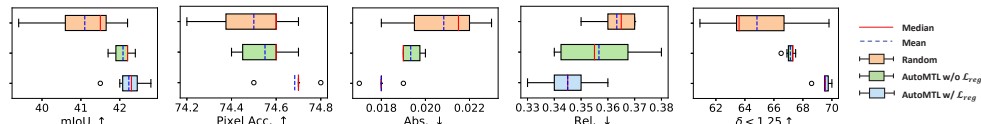

Figure 2: **Ablation Study on CityScapes.** The figures show the distributions of different evaluation metrics for three groups of multi-task models. The orange bar corresponds to the group of models generated from *Random* policies; the green and blue bars correspond to those sampled from the trained policy with or without the policy regularization (*AutoMTL w/o $\mathcal{L}_{reg}$* and *AutoMTL w/ $\mathcal{L}_{reg}$*).

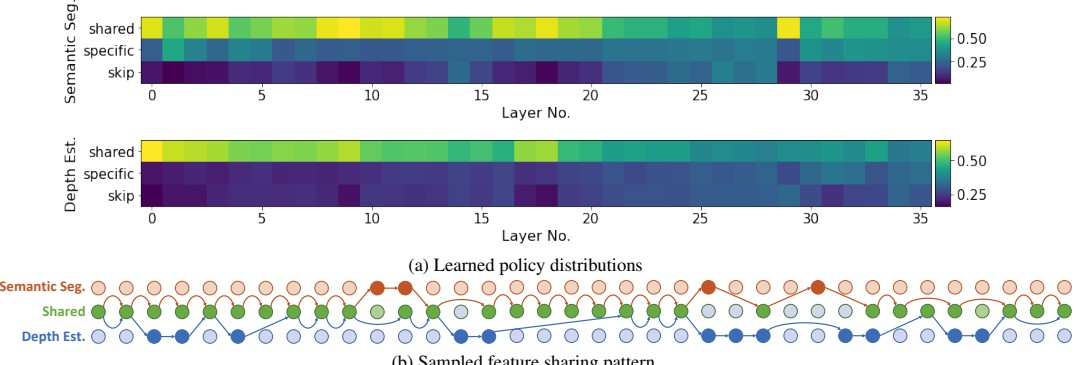

(a) Learned policy distributions

(b) Sampled feature sharing pattern

Figure 3: Policy Visualization for CityScapes.

and AdaShare have limited task performance improvement or even suffer from task performance degradation (see columns $\Delta t_2$ and $\Delta t_3$) on the Taskonomy dataset with five tasks. Similarly, the tree-like multi-task model search space in Learn to Branch limits the flexibility of the sharing patterns in its generated multi-task model, causing severe task interference as shown in columns $\Delta t_3$ to $\Delta t_5$ in Table 3. In contrast, AutoMTL could generate a multi-task model with larger capacity if necessary, leading to higher task performance, 13.6% higher than DEN, 3.1% than AdaShare, and 3.2% than Learn to Branch.

Furthermore, for semantic segmentation in NYUv2 (see columns $\Delta t_1$ in Table 2) and surface normal prediction in Taskonomy (see columns $\Delta t_2$ in Table 3), the performance of almost all the multi-task baselines are worse than the Single-Task baseline. It indicates that this particular task is negatively interfered by the other tasks when sharing parameters across them. In contrast, AutoMTL is still able to improve the performance of the two tasks because they tend to select more task-specific operators in our search space in order to reduce interference from the other tasks.

We also compare the inference time of different multi-task models in Table 4. AutoMTL could achieve a shorter inference time than independent models, soft-parameter sharing methods, and DEN. It is because AutoMTL allows some computation reuse when consecutive initial layers are shared among tasks and uses skip connections to further reduce the computation overhead. AutoMTL also achieves a competitive inference speed compared with AdaShare even though we provide additional task-specific options in the multi-task models.

Table 4: Inference Time (ms).

| Model | CityScapes (2 tasks) | NYUv2 (3 tasks) |
|---|---|---|
| Ind. Models | 71.01 | 107.65 |
| Multi-Task | 29.52 | 32.43 |
| Cross-Stitch | 71.01 | 107.65 |
| Sluice | 71.01 | 107.65 |
| NDDR | 67.41 | 101.89 |
| MTAN | 98.99 | 133.33 |
| DEN | 87.05 | 127.41 |
| AdaShare | 56.79 | 71.06 |
| AutoMTL | 60.84 | 80.41 |

The time cost of each training stage in terms of GPU hours is reported in Table 5. The experiments were conducted on an Nvidia RTX8000. Notice that the time cost of our architecture search process (the policy-train stage) is 12-13 GPU hours on CityScapes, 36-37 GPU hours on NYUv2, and about 120 GPU hours on Taskonomy, which are $13 \sim 133X$ faster than FBNetV5 [48] (1600 GPU hours on V100[2]), a state-of-the-art multi-task architecture search framework.

---

[2]Since the work is not open-sourced, we have no task performance comparison with it and the search costs here are extracted from the original paper. RTX8000 and V100 have similar computation capability and are hence comparable.

**Ablation Studies.** We present ablation studies to show the effectiveness of the architecture search process (the policy-train stage) and the proposed policy regularization term (Eq. 2). In Figure 2, we use a boxplot to show the distri-

Table 5: Time Cost of Training Stage (GPU hours).

| Stage | CityScapes | NYUv2 | Taskonomy |
|---|---|---|---|
| pre-train | 8-9 | 20-21 | ~25 |
| policy-train | 12-13 | 36-37 | ~120 |
| post-train | 14-15 | 44-45 | ~140 |

butions of different evaluation metrics for three groups of multi-task models. The orange group (*Random*) of models are generated from policies that are randomly initialized without policy-train, while the green (*AutoMTL w/o $\mathcal{L}_{reg}$*) and the blue (*AutoMTL w/ $\mathcal{L}_{reg}$*) groups are sampled from policies after the policy-train stage. The policy of the blue group is trained with the regularization but that of the green group is not. We generate six different models with seed $10 \sim 60$ in each group to compare their performance with less bias.

We make two main observations from Figure 2. First, both *AutoMTL w/o $\mathcal{L}_{reg}$* and *AutoMTL w/ $\mathcal{L}_{reg}$* achieve better task performance than *Random* in terms of the mean and the standard deviation of all the evaluation metrics. It indicates that the architecture search process is necessary and effective in predicting a good sharing pattern among tasks. Second, the mean of all metrics for *AutoMTL w/ $\mathcal{L}_{reg}$* is also better than *AutoMTL w/o $\mathcal{L}_{reg}$*, indicating that the proposed policy regularization term plays an important role in improving task performance. It echos the well-recognized benefits of parameter sharing among tasks in reducing overfitting and improving prediction accuracy.

To further illustrate the benefits of the proposed policy regularization term $\mathcal{L}_{reg}$, we provide more quantitative results on CityScapes with different $\lambda_{reg}$ in Table 6. We also list the Single-Task model and AdaShare as a comparison. All the other experiment settings including the training pipeline and the hyper-parameter setting remain the same.

Table 6: Quantitative Results on CityScapes with Different $\lambda_{reg}$. (Abs. Prf. & Rel. Prf.)

| Model | # Params ↓ | | Semantic Seg. | | | Depth Estimation | | | | | | $\Delta t$ ↑ |
|---|---|---|---|---|---|---|---|---|---|---|---|---|
| | Abs. (M) | Rel. (%) | mIoU ↑ | Pixel Acc. ↑ | $\Delta t_1$ ↑ | Error ↓ | | $\delta$, within ↑ | | | $\Delta t_2$ ↑ | |
| | | | | | | Abs. | Rel. | 1.25 | $1.25^2$ | $1.25^3$ | | |
| Single-Task | 42.569 | - | 36.5 | 73.8 | - | 0.026 | 0.38 | 57.5 | 76.9 | 87.0 | - | - |
| AdaShare | **21.285** | **-50.0** | 40.6 | 74.7 | +6.2 | **0.018** | 0.37 | **71.4** | 86.8 | 93.1 | +15.5 | +10.9 |
| $\lambda_{reg} = 0.01$ | **23.626** | **-44.5** | **43.4** | **74.9** | **+10.2** | 0.021 | 0.36 | 68.4 | 85.5 | 92.7 | +12.2 | +11.2 |
| $\lambda_{reg} = 0.001$ | 25.584 | -39.9 | 43.3 | 74.8 | +10.0 | 0.020 | 0.34 | **71.1** | **87.5** | **93.7** | +15.7 | +12.9 |
| $\lambda_{reg} = 0.0005$ | 28.819 | -32.3 | 42.8 | 74.8 | +9.3 | **0.018** | **0.33** | 70.0 | 86.6 | 93.4 | **+17.1** | **+13.2** |
| $\lambda_{reg} = 0.0001$ | 30.735 | -27.8 | 40.4 | 74.4 | +5.7 | 0.019 | 0.37 | 68.1 | 84.5 | 92.0 | +12.7 | +9.2 |

As the $\lambda_{reg}$ becomes larger, the probability of parameter sharing, especially those in initial layers, is higher, leading to the fewer number of parameters in the identified multi-task model but relatively lower task performance because of task interference. Users could adjust $\lambda_{reg}$ to control the tradeoff between resource efficiency and task accuracy. If the computation budget is limited, they could use a larger $\lambda_{reg}$ for a more compact model while sacrificing task accuracy. If the users call for the best task performance, they could tune $\lambda_{reg}$ to find the optimal setting.

**Policy Visualization.** We further visualize the learned sharing policies to reveal insights on the discovered multi-task architecture. Figure 3 shows the visualization of the learned policy distribution and the feature sharing pattern on CityScapes. For each layer in each task, Figure 3(a) illustrates its policy distribution $\pi$ introduced in Section 3.2. A brighter block indicates a higher probability of that operator being selected. The figure indicates that tasks tend to share bottom layers. Besides, Figure 3(b) provides a feature sharing pattern sampled from the learned policy distribution. The red arrows connect the operators used by semantic segmentation and the blue ones correspond to depth estimation. Operators that are not selected are semi-transparent. Overall, semantic segmentation is more likely to share operators with other tasks than depth estimation. Depth estimation has more than 25% of operators are skip connections, implying that this task prefers a more compact model than the backbone. Skip connections and operator sharing among the two tasks decrease the number of parameters in the multi-task model.

**Results on Other Backbone Models.** We also demonstrate the generality of AutoMTL by conducting experiments on CityScapes with two other typical backbone models MobileNetV2 [39] and MNasNet [45]. Without changing hyperparameters on this dataset, AutoMTL achieves 7.4% and 9.5% higher relative task performance with 33.5% and 35.9% fewer model parameters than the single-task baseline

and 6.5% and 11.1% higher relative task performance than the Multi-Task baseline for MobileNetV2 and MNasNet respectively.

**User Study on Ease-of-Use.** There is no re-implementation cost in AutoMTL when the backbone model changes. The compilation of a given backbone specified in prototxt format to a multi-task supermodel takes only ∼0.6s. On the contrary, users with proficient PyTorch skills in our user study still expect $20 \sim 40$ hours to complete re-implementation of the state-of-the-art NAS-based MTL approach Adashare [43].

## 5 Conclusion

In this work, we propose the first programming framework AutoMTL that generates compact multi-task models given an arbitrary input backbone CNN model and a set of tasks. AutoMTL features a multi-task supermodel compiler that automatically transforms any given backbone CNN into a multi-task supermodel that encodes the proposed stretchable architecture search space. Then through policy approximation and regularization, the architecture search component effectively identifies good sharing policies that lead to both high task accuracy and memory efficiency. Experiments on three popular multi-task learning benchmarks demonstrate the superiority of AutoMTL over state-of-the-art approaches in terms of task accuracy and model size.

**Limitations and Broader Impact Statement.** Our research facilitates the adoption of multi-task learning techniques to solve many tasks at once in resource-constraint scenarios. Particularly, we offer the first systematic support for automating efficient multi-task model development for vision tasks. The support of other AI tasks (e.g., NLP tasks) is left as future work. It has a positive impact on applications that tackle multiple tasks such as environment perceptions for autonomous vehicles and human-computer interactions in robotic, mobile, and IoT applications. The negative social impact of our research is difficult to predict since it shares the same pitfalls with general deep learning techniques that suffer from dataset bias, adversarial attacks, fairness, etc.

**Acknowledgement.** This work is supported by UMass Amherst Start-up Funding and Adobe Research Collaboration Grant.

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
