# A Compiling Procedure

MTS-Compiler takes as inputs a user-specified backbone model in the format of prototxt and a task list, and then generates a multi-task supermodel represented as a graph of VCNs. Algorithm 1 elaborates the compiling procedure. The *backbone* is first parsed into a list of operators *ops* (line 4). Then the compiler will iterate over *ops* to initialize the corresponding VCNs (line 5-7). The final multi-task supermodel *mtSuper* is a list of VCNs.

---
**Algorithm 1** Compiling Procedure
---
1: **Input:** *backbone*: a backbone model in prototxt format; *tasks*: the IDs of tasks to learn.
2: **Output:** *mtSuper*: a multi-task supermodel
3: $mtSuper = [\,]$
4: *ops* = parse_prototxt(*backbone*)
5: **for** *op* in *ops* **do**
6:    *mtSuper*.append(VCN(*op*, *tasks*))
7: **end for**
8:
9: **Class** VCN:
10: **function** init(*op*, *tasks*)
11:    $self.op = op$
12:    $self.parents = [\,]$
13:    **for** *p* in *op.parentOps* **do**
14:       $self.parents$.append(getVCN(*p*))
15:    **end for**
16:    **for** *t* in *tasks* **do**
17:       $self.sp^t = op$.deepcopy()
18:       $self.sk^t$ = SkipConnection()
19:       $self.\mathcal{P}^t$ = Gumbel-Softmax([0., 0., 0.])
20:    **end for**
21: **end function**
---

# B Hyper-parameter Settings

Table 5 summarizes the hyper-parameters used in training. For CityScapes and NYUv2, AutoMTL spends 10,000 iterations on pre-training the supermodel (pre-train), then 20,000 iterations for training policy and supermodel jointly (policy-train), and finally 30,000 iterations for training the identified multi-task model (post-train). As for Tiny-Taskonomy, the three stages need 20,000, 30,000, 50,000 iterations respectively to converge. The hyper-parameters are chosen by empirical experience in AdaShare [44] and manual search during our experiments.

Table 5: Hyper-parameters for training CityScapes, NYUv2, and Tiny-Taskonomy.

| Dataset | weight lr | policy lr | weight lr decay | $\lambda_{seg}$ | $\lambda_{sn}$ | $\lambda_{depth}$ | $\lambda_{kp}$ | $\lambda_{edge}$ | $\lambda_{reg}$ |
|---|---|---|---|---|---|---|---|---|---|
| CityScapes | 0.001 | 0.01 | 0.5/4,000 iters | 1 | - | 1 | - | - | 0.0005 |
| NYUv2 | 0.001 | 0.01 | 0.5/4,000 iters | 5 | 20 | 5 | - | - | 0.001 |
| Tiny-Taskonomy | 0.0001 | 0.01 | 0.3/10,000 iters | 1 | 3 | 2 | 7 | 7 | 0.0005 |

# C Full Comparison of All Metrics on NYUv2 and Taskonomy

The full comparison of all metrics on NYUv2 and Taskonomy are summarized in Table 6 and 7. On NYUv2, AutoMTL could achieve outstanding performance on 7 out of 12 metrics, while on Taskonomy, AutoMTL outperforms all the baselines on almost all the 5 metrics.

# D Ablation Studies on NYUv2

The ablation studies are also conducted on NYUv2. The same phenomenon as Section 4.2 Ablation Studies in the main paper can be observed in Figure 4. In short, both the architecture search process and the proposed policy regularization term are indispensable to obtain a feature-sharing pattern with high task performance.

Table 6: Quantitative results on NYUv2. (Abs. Prf.)

| Model | # Params (M) ↓ | Semantic Seg. | | Surface Normal Prediction | | | | | Depth Estimation | | | | |
|---|---|---|---|---|---|---|---|---|---|---|---|---|---|
| | | mIoU ↑ | Pixel Acc. ↑ | Error ↓ | | $\theta$, within ↑ | | | Error ↓ | | $\delta$, within ↑ | | |
| | | | | Mean | Median | 11.25° | 22.5° | 30° | Abs. | Rel. | 1.25 | $1.25^2$ | $1.25^3$ |
| Single-Task | 63.855 | 26.5 | **58.2** | 17.7 | 16.3 | 29.4 | **72.3** | **87.3** | 0.62 | 0.24 | 57.8 | 85.8 | 96.0 |
| Multi-Task | 21.285 | 22.2 | 54.4 | 17.2 | 15.8 | 32.2 | 70.5 | 84.8 | 0.59 | 0.22 | 60.9 | 87.7 | 96.7 |
| Cross-Stitch | 63.855 | 25.4 | 57.6 | 17.2 | 14.0 | 41.4 | 67.7 | 80.4 | 0.58 | 0.23 | 61.4 | 88.4 | 95.5 |
| Sluice | 63.855 | 23.8 | 56.9 | 17.2 | 14.4 | 38.9 | 69.0 | 81.4 | 0.58 | 0.24 | 61.9 | 88.1 | 96.3 |
| NDDR-CNN | 67.047 | 21.6 | 53.9 | **17.1** | 14.5 | 37.4 | 70.9 | 83.1 | 0.66 | 0.26 | 55.7 | 83.7 | 94.8 |
| MTAN | 66.217 | 26.0 | 57.2 | 17.2 | 13.9 | **43.7** | 70.5 | 81.9 | 0.57 | 0.25 | 62.7 | 87.7 | 95.9 |
| DEN | 23.838 | 23.9 | 54.9 | **17.1** | 14.8 | 36.0 | 70.6 | 83.4 | 0.97 | 0.31 | 22.8 | 62.4 | 88.2 |
| AdaShare | 21.285 | 24.4 | 57.8 | 17.7 | **13.8** | 42.3 | 68.9 | 80.5 | 0.59 | **0.20** | 61.3 | 88.5 | 96.5 |
| AutoMTL | 35.056 | **26.6** | **58.2** | 17.3 | 14.4 | 39.1 | 70.7 | 83.1 | **0.54** | 0.22 | **65.1** | **89.2** | **96.9** |

Table 7: Quantitative results on Taskonomy. (Abs. Prf.)

| Models | # Params (M) ↓ | Semantic Seg. ↓ | Surface Normal ↑ | Depth Est. ↓ | Keypoint Det. ↓ | Edge Det. ↓ |
|---|---|---|---|---|---|---|
| Single-Task | 106.424 | 0.575 | 0.807 | **0.022** | 0.197 | 0.212 |
| Multi-Task | 21.285 | 0.596 | 0.796 | 0.023 | 0.197 | 0.203 |
| Cross-Stitch | 106.424 | 0.570 | 0.779 | **0.022** | 0.199 | 0.217 |
| Sluice | 106.424 | 0.596 | 0.795 | 0.024 | 0.196 | 0.207 |
| NDDR-CNN | 115.151 | 0.599 | 0.800 | 0.023 | 0.196 | 0.203 |
| MTAN | 95.994 | 0.621 | 0.787 | 0.023 | 0.197 | 0.206 |
| DEN | 23.838 | 0.737 | 0.786 | 0.027 | 0.192 | 0.203 |
| AdaShare | 21.285 | 0.562 | 0.802 | 0.023 | **0.191** | 0.200 |
| Learn to Branch | 30.650 | **0.521** | 0.850 | 0.023 | 0.202 | 0.217 |
| AutoMTL | 53.106 | 0.558 | **0.873** | **0.022** | **0.191** | **0.197** |

# E    Ablation Study on the Training Pipeline

We also conducted additional ablation study on the proposed three-stage training pipeline. Specifically, the necessity of the policy-train stage is verified in Section 4.2 already, so we focus on the pre-train and the post-train stages in this section.

The quantitative results on CityScapes with and without the pre-train stage are shown in Table 8. The results are collected using the same hyper-parameter setting reported in the paper. We can see that AutoMTL with pre-training can obtain higher task performance than the model without pre-training. This observation echoes existing work in NAS [54], which also suggests warming up the supermodel first and then conducting searching. Both ablation studies in [54] and our empirical study demonstrate that such a pre-train stage produces a better initialization for the parameters of the supermodel and eventually results in a more accurate multi-task architecture with a similar amount of parameters.

The post-train stage can either use fine-tuning or training-from-scratch. Table 9 compares the task performance of the two options on the identical sampled multi-task architecture under the same hyper-parameter setting. The results show that re-training the identified multi-task model from scratch would produce higher task performance, which suggests retraining as a better post-train strategy. The observation is consistent with that of the well-known differentiable NAS method DARTS [30], which has become a common practice in recent years [49, 54, 53]. Note that in our paper, we also retrain all baselines from scratch for a fair comparison.

# F    Policy Visualizations on NYUv2 and Taskonomy

Figure 5 visualizes the learned policy distribution on NYUv2. It can be seen that for top layers near the output, the semantic segmentation and the depth estimation tend to have their own computation operators instead of sharing with other tasks. This trend leads that the two tasks may suffer less from the negative interference between tasks, which becomes a possible explanation of why the model searched by AutoMTL can have better task performance on them than existing methods as analyzed in Section 6.2 in the main paper.

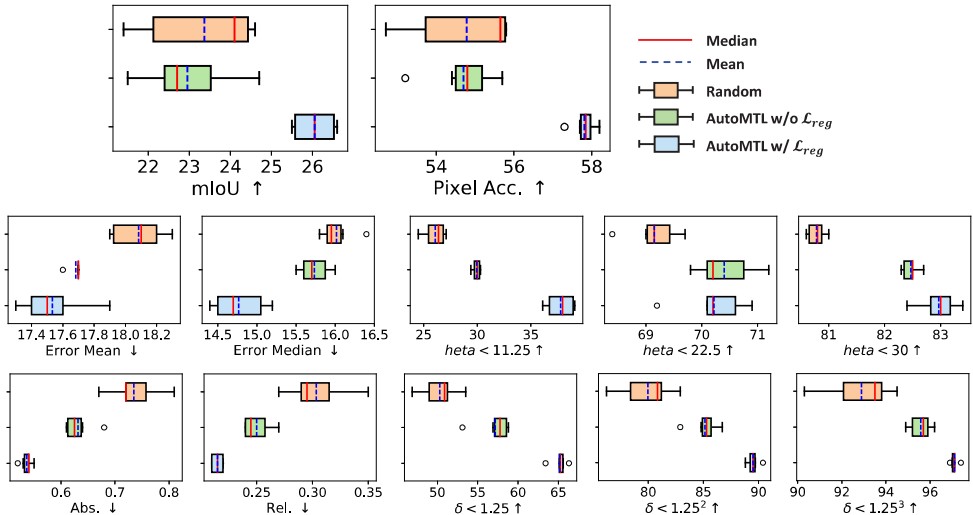

Figure 4: Ablation study on NYUv2. Distributions of different metrics for three groups of multi-task models are exhibited, including models generated from *Random* policies or those sampled from the trained policy with or without the policy regularization (*AutoMTL w/o $\mathcal{L}_{reg}$* and *AutoMTL w/ $\mathcal{L}_{reg}$*).

Table 8: Ablation study about the pre-train stage on CityScapes.

| Model | # Params ↓ | | Semantic Seg. | | | Depth Estimation | | | | | | Δt ↑ |
|---|---|---|---|---|---|---|---|---|---|---|---|---|
| | Abs. (M) | Rel. (%) | mIoU ↑ | Pixel Acc. ↑ | $\Delta t_1$ ↑ | Error ↓ | | δ, within ↑ | | | $\Delta t_2$ ↑ | |
| | | | | | | Abs. | Rel. | 1.25 | $1.25^2$ | $1.25^3$ | | |
| Single-Task | 42.569 | - | 36.5 | 73.8 | - | 0.026 | 0.38 | 57.5 | 76.9 | 87.0 | - | - |
| AutoMTL w/o pre-train | **28.878** | **-32.1** | 41.1 | 74.5 | +6.8 | 0.020 | 0.41 | 65.7 | 82.7 | 90.6 | +8.2 | +7.5 |
| AutoMTL w/ pre-train | 30.096 | -29.3 | **42.8** | **74.8** | **+9.3** | **0.018** | **0.33** | **70.0** | **86.6** | **93.4** | **+17.1** | **+13.2** |

Figure 6 visualizes the learned policy distribution on Taskonomy. By comparing the brightness of the three branches in each layer (brighter means higher probability to be chosen), it can be found that the brightness differences between the three branches are more salient in layer No. 0∼10 and No. 30∼35, which indicates that tasks would be more likely to have branch preferences in the top and bottom layers. While for intermediate layers (layer No. 18∼28), the chance of each branch being selected is basically equal. This phenomenon is consistent with traditional multi-task model design principles, which pay more attention to top and bottom layers to decide whether they should be shared or not.

# G  Task Correlation

We use cosine similarity between task-specific policies to quantify task correlations. Figure 7 illustrates the task correlations in Taskonomy (the darker the higher correlation) and we can make the following observations. (a) Semantic segmentation has a relatively weak correlation with depth estimation compared to other tasks. (b) Surface normal prediction has good correlations with all the other tasks. (c) Depth estimation has low correlations with keypoint and edge detection. (d) Keypoint detection has a strong correlation with edge detection.

# H  Extension to other Architectures

The users are able to feed any convolution-based model into the proposed MTS-Compiler. We try to demonstrate the superiority of this function by quantifying the manual efforts of using AutoMTL or AdaShare [44] when users switch to different backbones. Specifically, we invited 7 graduate students who are proficient in PyTorch and Machine Learning to convert the backbone model from Deeplab-ResNet34 to MobileNetV2 in both AutoMTL and AdaShare and then record their working time. When using AutoMTL, they firstly spent around 20 minutes reading through our document. After that, all they need to do is to download a MobileNetV2 prototxt from the Internet and use our MTS-Compiler and trainer tools directly. On the other hand, since the public implementation of AdaShare is based on Deeplab-ResNet34, our participants had to re-implement MobileNetV2 as well as

Table 9: Ablation study about the post-train stage on CityScapes.

| Model | # Params ↓ | | Semantic Seg. | | | Depth Estimation | | | | | | $\Delta t$ ↑ |
|---|---|---|---|---|---|---|---|---|---|---|---|---|
| | Abs. (M) | Rel. (%) | mIoU ↑ | Pixel Acc. ↑ | $\Delta t_1$ ↑ | Error ↓ | | $\delta$, within ↑ | | | $\Delta t_2$ ↑ | |
| | | | | | | Abs. | Rel. | 1.25 | $1.25^2$ | $1.25^3$ | | |
| Single-Task | 42.569 | - | 36.5 | 73.8 | - | 0.026 | 0.38 | 57.5 | 76.9 | 87.0 | - | - |
| post-train w/ fine-tune | 30.096 | -29.3 | 41.8 | 74.6 | +7.8 | 0.019 | 0.38 | 68.1 | 84.8 | 92.2 | +12.3 | +10.1 |
| post-train w/ retrain | 30.096 | -29.3 | **42.8** | **74.8** | **+9.3** | **0.018** | **0.35** | **69.7** | **85.7** | **92.9** | **+15.6** | **+12.5** |

the embedded policy from scratch to fit into the AdaShare training framework. According to the feedback, it generally took $20 \sim 40$ hours to complete coding and debugging.

We also conduct experiments on CityScapes with two other typical backbone models MobileNetV2 [40] and MNasNet [46]. Table 10 and 11 report the task performance when constructing multi-task models on them. AutoMTL always could search out a better multi-task architecture when compared to the vanilla multi-task model.

Table 10: Quantitative results with MobileNetV2 on CityScapes.

| Model | # Params (%) ↓ | Semantic Seg. | | | Depth Estimation | | | | | | $\Delta t$ ↑ |
|---|---|---|---|---|---|---|---|---|---|---|---|
| | | mIoU ↑ | Pixel Acc. ↑ | $\Delta t_1$ ↑ | Error ↓ | | $\delta$, within ↑ | | | $\Delta t_2$ ↑ | |
| | | | | | Abs. | Rel. | 1.25 | $1.25^2$ | $1.25^3$ | | |
| Single-Task | - | 25.9 | 63.5 | - | 0.043 | 0.53 | 32.1 | 71.1 | 86.5 | - | - |
| Multi-Task | -50.0 | 26.3 | 63.4 | +0.7 | 0.042 | 0.48 | 33.6 | 66.5 | 82.9 | +1.2 | +0.9 |
| AdaShare | -50.0 | **26.7** | 61.2 | -0.3 | **0.032** | **0.46** | 42.1 | 71.6 | 84.3 | +13.6 | +6.7 |
| AutoMTL | -33.5 | 25.8 | **63.7** | +0.0 | 0.035 | 0.47 | **44.4** | **74.8** | **87.3** | **+14.9** | **+7.4** |

Table 11: Quantitative results with MNasNet on CityScapes.

| Model | # Params (%) ↓ | Semantic Seg. | | | Depth Estimation | | | | | | $\Delta t$ ↑ |
|---|---|---|---|---|---|---|---|---|---|---|---|
| | | mIoU ↑ | Pixel Acc. ↑ | $\Delta t_1$ ↑ | Error ↓ | | $\delta$, within ↑ | | | $\Delta t_2$ ↑ | |
| | | | | | Abs. | Rel. | 1.25 | $1.25^2$ | $1.25^3$ | | |
| Single-Task | - | **25.5** | 63.6 | - | 0.040 | 0.49 | 36.7 | 73.3 | 87.3 | - | - |
| Multi-Task | -50.0 | 25.1 | 63.5 | -0.9 | 0.040 | 0.48 | 35.9 | 67.8 | 84.2 | -2.2 | -1.6 |
| AutoMTL | -35.9 | **25.5** | **63.7** | **+0.1** | **0.028** | **0.43** | **53.7** | **77.1** | **87.8** | **+18.9** | **+9.5** |

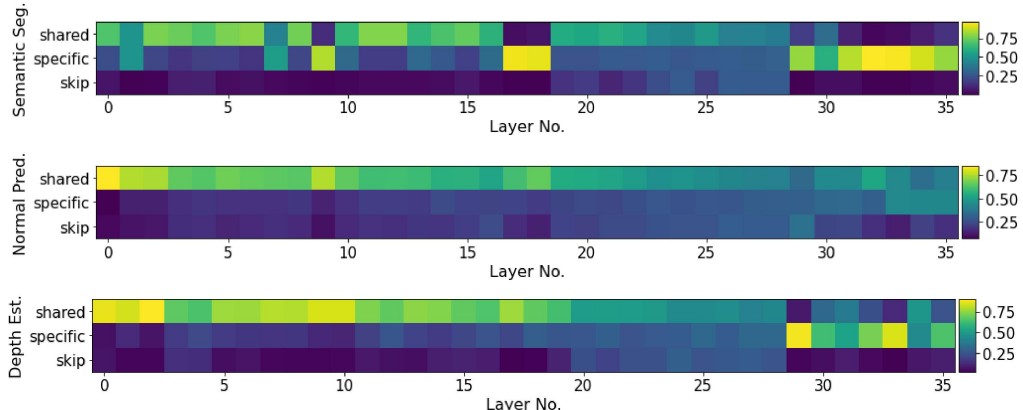

Figure 5: Learned policy distributions for the three tasks in NYUv2.

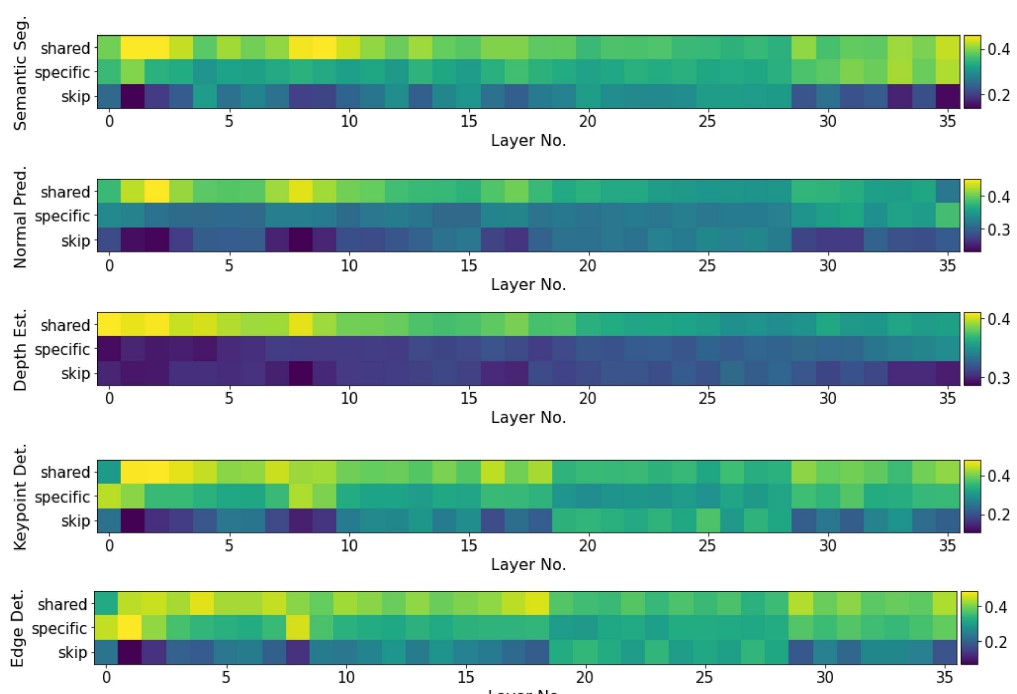

Figure 6: Learned policy distributions for the five tasks in Taskonomy.

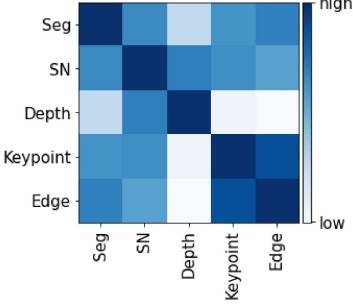

Figure 7: Task correlations in Taskonomy.