# OpenReview forum: "AutoMTL: A Programming Framework for Automating Efficient Multi-Task Learning"
_NeurIPS.cc/2022/Conference — NeurIPS 2022 Accept_

### Official Review · Reviewer_yPp6 · 2022-07-10

**Rating:** 5
**Confidence:** 4
**Soundness:** 2 fair
**Presentation:** 3 good
**Contribution:** 2 fair

**Summary:**

The paper focuses automatically learning to produce efficient neural network architectures for jointly performing multiple tasks given an arbitrary backbone and a set of tasks. The proposed method can efficiently generate efficient network for multi-task learning and achieve good performance on three multi-task learning benchmarks.

**Questions:**

1. The idea of learning multi-task architecture is very similar to Adashare while the proposed method has one more operation in each layer (task-specific) such that in a layer each task can keep the layer task-specific (Adashare only have skip or not which make the layer shared across tasks or shared by a subset of tasks or skipped by all tasks). It enhance the power of learning more powerful representations however, it has a larger action space to learn and it can be computationally costly. In this point of view, the novelty of this work is limited.

2. One of the challenge in multi-task learning is the optimization of the multi-task learning network, which is not discussed in this paper and need to be discussed. The optimization of multi-task learning strategies, such as loss weighting, knowledge distillation strategies and gradient-based methods can greatly affects the performance of multi-task learning. Is the optimization of multi-task learning also affects the the network architecture learning and the training of the final network?

3. As mentioned above, the optimization can greatly affects the performance of multi-task learning and it may differs for different architectures. Do the authors tune the hyper-parameters lambda (losses weight) for the proposed method and apply them to other methods? It can be more clear that if all network learned with uniform loss weights or also tune the loss weights for different architectures.

4. The proposed method has more parameters (around +20%) than Adashre while the gain is not significant. The performance gains on NYUv2 dataset and Taskonomy are not significant (Table 6 and Table 7 in appendix). It also looks strange that MTAN uses much more parameters than the vanilla multi-task learning baseline in all datasets and single-task learning in NYUv2. The authors are suggested to check again this.

5. The when applying the proposed method using MobileNetV2 and MNasNet, only Vanilla Multi-task learning baseline and single-task learning baseline are compared, how well are other methods?

**Limitations:**

The limitation of methods and potential negative societal impact are discussed in the paper.

**Strengths And Weaknesses:**

Strengths:
1. The paper develops an efficient programming framework that can automatically learn to produce efficient neural network architectures for jointly performing multiple tasks given an arbitrary backbone and a set of tasks.

2. It is shown that the method performs better than related multi-task architecture learning approaches in most metrics in three datasets.

3. And the authors also show that the proposed framework is easier than related literatures e.g. Adashare, for producing multi-task learning network

Weaknesses:

1. The main weaknesses of the paper is its limited novelty as the idea is quite similar to Adashare and its high computational cost for searching the multi-task learning architectures. (Detailed in Questions section.)

2. Another limitation is the effect of the optimization in multi-task learning is not considered while the optimization in multi-task learning can greatly influence the performance of multi-task learning (See Question section for more detailed).

3. The performance gains are not significant while the method requires more parameters than Adashare. (See Questions section for more detailed.)

---

> ### Author Response · Authors · 2022-08-02
> **Responses to Reviewer yPp6**
>
> Thank you for your constructive comments!
>
> **Q1: The novelty compared with AdaShare.**
>
> **A1**: We agree with the reviewer that in terms of search space, we extend the search space of AdaShare by offering task-specific options for each task. The larger search space allows AutoMTL to get rid of the limitations of the backbone capacity and identify better multi-task models.
>
> **The most important feature that differentiates AutoMTL from existing methods like AdaShare is that we propose an easy-to-use programming framework for automating resource-efficient multi-task model development.** Specifically, AutoMTL has three major superiorities:
>
> (1) AutoMTL could **automatically generalize to different backbone models without re-implementation efforts.** One can switch to any CNN model by simply providing its architecture prototxt. Existing works, however, need to be re-implemented for different backbone models because their implementations are highly coupled with the backbone model.
> For example, as shown in our user study on easy-of-use (Line 360-364) and Appendix Section H (original Section G), **the re-implementation effort for AdaShare is non-negligible (20-40 hours) for proficient ML users, while AutoMTL takes only ~0.6s for auto-recompilation when the backbone model changes.**
>
> (2) AutoMTL is **the first MTL infrastructure** that seamlessly integrates the MTS-Compiler, a set of PyTorch-based APIs, the architecture search algorithm, and a training pipeline. Users are free to either use APIs or feed the backbone pre-defined prototxt to the MTS-Compiler in order to build the MTL supermodel. Besides, with the integrated training pipeline, users can automatically identify the efficient multi-task model.
> We do hope the community could embrace novelties not only in learning algorithms, but also in ML infrastructures and programming frameworks. Significant advances in the domain might need the synergy of both novelties.
>
> (3) AutoMTL **considers parameter sharing at the operator level** rather than the block level as in prior works such as Learn-to-Branch and AdaShare. The operator-level sharing granularity not only **enables the automatic support of arbitrary CNN backbone architectures**, but also **leads to a stretchable architecture search space that contains multi-task models with a wide range of model capacity**. Thanks to gradient-based search, the larger search space doesn’t increase computation cost.
>
> **Q2: The searching cost compared with AdaShare.**
>
> **A2**: In terms of the searching time (GPU hours on Nvidia RTX8000), AutoMTL doesn’t require a significantly higher cost compared with AdaShare. The table below reports the time to train the supermodel on three benchmarks for both Adashare and AutoMTL. Please refer to our comments to **“Q2: How long does it take to train the model?” of Reviewer koYc** for more results.
>
> |   Model  | CityScapes | NYUv2 | Taskonomy |
> |:--------:|:----------:|:-----:|:---------:|
> | AdaShare |    9-10    |  ~35  |    ~111   |
> |  AutoMTL |    12-13   | 36-37 |    ~120   |
>
> **Q3: About the optimization in MTL.**
>
> **A3**: We agree with you that one of the challenges in MTL is the optimization of MTL models. This research direction is, however, **orthogonal to the multi-task architecture design**. Several works along this line (e.g., Auto-λ [1], CAGrad [2], and PCGrad [3]) propose new optimization techniques and validate their optimizers on established multi-task model designs (e.g., multi-task baseline).
>
> On the contrary, our paper as well as prior works such as AdaShare, DEN, and Learn-to-Branch, etc. focus on the architecture design and use established neural network optimization methods, (e.g., weighted loss with Adam optimizer).  As the reviewer suggested, it could be an interesting empirical study (in the future) to investigate how various optimizers affect the architecture searching process and the performance of multi-task models. This study is not the focus of our work.
>
> **Note that, on top of our AutoMTL framework, users are free to use existing optimization methods to further improve the task performance of multi-task models.**
>
> **Q4: Do the authors tune the hyper-parameters lambda (losses weight)?**
>
> **A4**: Yes, we tune the hyper-parameters manually as reported in Appendix Section B, and use the same parameter set for our model and baselines.
>
> [1] Liu, S., James, S., Davison, A. J., & Johns, E. Auto-Lambda: Disentangling Dynamic Task Relationships.TMLR 2022.
> [2] Liu, B., Liu, X., Jin, X., Stone, P., & Liu, Q. Conflict-Averse Gradient Descent for Multi-task Learning. NeurIPS 2021.
> [3] Yu, T., Kumar, S., Gupta, A., Levine, S., Hausman, K., & Finn, C. Gradient Surgery for Multi-Task Learning. NeurIPS 2020.

---

> > ### Author Response · Authors · 2022-08-02
> > **Responses to Reviewer yPp6**
> >
> > **Q5: The performance gain compared with AdaShare.**
> >
> > **A5**: Both AdaShare and our paper aim at optimizing task accuracy and model size. The reason our approach results in more parameters than Adashare is that **we target a higher task accuracy not only averaged over all tasks, but also for each task individually in an implicit way. AutoMTL’s search space provides the capability for each task to improve/preserve its accuracy by owning more task-specific parameters if necessary**. It is also the reason why AutoMTL provides better overall task accuracy averaged over all tasks.
> >
> > For example, on NYUv2, AdaShare suffers from a 4.3% accuracy drop on Semantic Segmentation (t1 in Table 2) while AutoMTL can still provide a 0.2% accuracy improvement on the same task by allowing this task to own more task-specific parameters. For a similar reason, on Taskonomy, AdaShare suffers from a 0.6% and 4.5% accuracy drop on Surface Normal and Depth Estimation (t2 and t3 in Table 3) while AutoMTL still achieves 8.2% and 0% accuracy improvement on the same two tasks respectively.
> >
> > Besides, our experiment results shown below also demonstrate that, even with a similar amount of model parameters, AutoMTL can achieve better overall task performance averaged across tasks. **In practice, users could adjust the hyper-parameter $\lambda_{reg}$ to balance the tradeoff between model size  (memory footprint) and task accuracy.**
> >
> > |         Model         | # Params (M) |  Seg. mIoU $\uparrow$ |  Seg.  Pixel Acc. $\uparrow$        |    $\Delta t_1 \uparrow$                   | Depth Abs. Error  $\downarrow$ |  Depth Rel. Error  $\downarrow$    | Depth  $\delta < 1.25 \uparrow$       |    Depth $\delta < 1.25^2 \uparrow$    |   Depth  $\delta < 1.25^3 \uparrow$    |     $\Delta t_2 \uparrow$                   | $\Delta t \uparrow$ |
> > |:---------------------:|:------------:|:---------------------:|:----------:|:---------------------:|:----------------:|:----:|:-------:|:------:|:------:|:---------------------:|:------------------:|
> > |    Single-Task   |    -   |   36.5   |   73.8   |     -     |   0.026   |   0.38   |   57.5   |   76.9   |   87.0   |     -     |     -     |
> > |     AdaShare     | 21.285 |   40.6   |   74.7   |    +6.2   | **0.018** |   0.37   | **71.4** |   86.8   |   93.1   |   +15.5   |   +10.9   |
> > | AutoMTL  w/ $\lambda_{reg}=0.01$ | 23.626 | **43.4** | **74.9** | **+10.2** |   0.021   |   0.36   |   68.4   |   85.5   |   92.7   |   +12.2   |   +11.2   |
> > | AutoMTL w/ $\lambda_{reg}=0.001$ | 25.584 |   43.3   |   74.8   |   +10.0   |   0.020   | **0.34** |   71.1   | **87.5** | **93.7** | **+15.7** | **+12.9** |
> >
> > **Q6: The number of parameters in MTAN and the comparison with the multi-task baseline.**
> >
> > **A6**: **Compared to the vanilla multi-task model, none of the other baselines, not alone MTAN, can compete with the multi-task baseline in terms of inference time and model size.** This is because, in the multi-task baseline, all the parameters in the backbone are shared across tasks. Thus, the multi-task baseline always achieves the maximum reduction in inference time and model size. However, the multi-task baseline frequently suffers from poor task accuracy, even worse than independent models.
> >
> > Thank you for your suggestion of re-checking the number of parameters in MTAN. We do figure out that the relative number of parameters in MTAN on CityScapes should be revised to +20.5% rather than -20.5%. **In other words, MTAN has more parameters even compared to the single-task baseline because of task-specific attention modules.** Specifically, each attention module would introduce ~15M additional parameters to the backbone model with 21M parameters, leading to +20.5% parameter growth on CityScapes with 2 tasks (21+15x2=51M) and +3.7% on NYUv2 with 3 tasks (21+15x3=66M), when compared to the single-task baseline.
> >
> > **Q7: When using MobileNetV2 and MNasNet, only Vanilla Multi-task learning baseline and single-task learning baseline are compared.**
> >
> > **A7**: The experiments on MobileNetV2 and MNasNet in Appendix Section H (original Section G) aim to demonstrate **the generality of AutoMTL to an arbitrary CNN backbone without re-implementation efforts.** It is very time-consuming and difficult to implement other baselines on different backbones since their implementations are deeply coupled with a specific CNN model. This is exactly the reason that motivates us to develop the AutoMTL framework that can automatically support any CNN architecture without reimplementation efforts. For example, according to our user study (Line 360-364), users with proficient PyTorch skills still expect 20 ∼ 40 hours to re-implement AdaShare, while AutoMTL takes only ~0.6s to complete auto-recompilation.

---

> > > ### Comment · Reviewer_yPp6 · 2022-08-09
> > > **Thank you for your response**
> > >
> > > Thank you for your response and most of my concerns are addressed. However, the related work section is not satisfying. Some related literatures [1-4] are missing and the authors are suggested to discuss the literatures of MTL optimization [5-11] and task grouping [12,13] as they are quite related and would potentially affect or be complementary to the proposed method in the paper. I will raise my score.
> > >
> > > [1] Vandenhende, Simon, et al. "Multi-task learning for dense prediction tasks: A survey." TPAMI, 2021.
> > >
> > > [2] Zhang, Yu, and Qiang Yang. "A survey on multi-task learning." TKDE, 2021.
> > >
> > > [3] Vandenhende, Simon, et al. "Branched multi-task networks: deciding what layers to share." BMVC, 2020.
> > >
> > > [4] Bragman, Felix JS, et al. "Stochastic filter groups for multi-task cnns: Learning specialist and generalist convolution kernels." ICCV, 2019.
> > >
> > > [5] Sener, Ozan, and Vladlen Koltun. "Multi-task learning as multi-objective optimization." Neurips, 2018.
> > >
> > > [6] Kendall, Alex, Yarin Gal, and Roberto Cipolla. "Multi-task learning using uncertainty to weigh losses for scene geometry and semantics." CVPR, 2018.
> > >
> > > [7] Chen, Zhao, et al. "Gradnorm: Gradient normalization for adaptive loss balancing in deep multitask networks." ICML, 2018.
> > >
> > > [8] Yu, Tianhe, et al. "Gradient surgery for multi-task learning." Neurips, 2020.
> > >
> > > [9] Li, Wei-Hong, and Hakan Bilen. "Knowledge distillation for multi-task learning." ECCV Workshop, 2020.
> > >
> > > [10] Liu, Bo, et al. "Conflict-averse gradient descent for multi-task learning." Neurips, 2021.
> > >
> > > [11] Liu, Shikun, et al. "Auto-Lambda: Disentangling Dynamic Task Relationships." TMLR, 2022.
> > >
> > > [12] Standley, Trevor, et al. "Which tasks should be learned together in multi-task learning?." ICML, 2020.
> > >
> > > [13] Fifty, Chris, et al. "Efficiently identifying task groupings for multi-task learning." Neurips, 2021.

---

> > > > ### Author Response · Authors · 2022-08-09
> > > > **Related Work Revision**
> > > >
> > > > Thank you for your suggestions! We're revised our related work section to include more literatures and one part for MTL optimization methods. The new version of our submission has been uploaded.

---

> ### Author Response · Authors · 2022-08-08
> **Look Forward to Discussing with You**
>
> Dear reviewer yPp6:
>
> We sincerely thank you for the review and comments. We have provided corresponding responses and results, which we believe have covered your concerns. We hope to further discuss with you whether or not your concerns have been addressed. Please let us know if you still have any unclear parts of our work.
>
> Best,
> Authors of Paper 6905

---

> ### Author Response · Authors · 2022-08-08
> **The Last 24 hours for Discussion**
>
> Dear reviewer yPp6:
>
> We have provided corresponding responses and results, which we believe have covered your concerns. Notice that there are less than 24 hours left for the discussion period. We really hope to confirm with you whether or not your concerns have been addressed. Thank you very much!
>
> Best,
> Authors of Paper 6905

---

### Official Review · Reviewer_koYc · 2022-07-11

**Rating:** 6
**Confidence:** 4
**Soundness:** 3 good
**Presentation:** 3 good
**Contribution:** 3 good

**Summary:**

The AutoMTL system integrates the Multi-Task Supermodel Compiler (MTS-Compiler), which transforms a user-provided backbone CNN into a multi-task supermodel that encodes the architecture search space. The authors propose a Stretchable Architecture Search Space that offers flexibility in deriving multi-task models with a wide range of model capacity based on task difficulties and interference, and further propose a novel data structure called Virtual Computation Node to encode the search space and enable compiler-based multi-task supermodel transformation. They propose a policy regularization term on the sharing policy to promote parameter sharing for high memory efficiency.

**Questions:**

How long does it take to train the model? It seems that the training pipeline is complicated. It would be better to show the comparison of training time among baselines.

**Limitations:**

The method is interesting and the model can be adapted to other backbone without implementation cost. However, it seems that the training time and inference time is longer than the baseline. The method cannot achieve best on all evaluation metrics.

**Strengths And Weaknesses:**

Strengths:
1. The paper proposes a  resource-efficient architecture design, which determines what parameters of a backbone model to share across tasks to optimize for both resource efficiency and task accuracy.
2. AutoMTL takes as inputs an arbitrary backbone CNN and a set of tasks to learn, and then produces a multi-task model that achieves high task accuracy and small memory footprint.
3. This method considers parameter sharing in MTL at the operator level. The operator-level sharing granularity not only enables the automatic support of arbitrary CNN backbone architectures, but also leads to a stretchable architecture search space that contains multi-task models with a wide range of model capacity.

Weaknesses:
1. Even though AutoMTL achieves better results, the number of model parameters is larger than Multi-Task baseline.
2. The inference time of AutoMTL in CityScapes is longer than Multi-Task and AdaShare.

---

> ### Author Response · Authors · 2022-08-02
> **Responses to Reviewer koYc**
>
> Thank you for your positive feedback!
>
> **Q1: Even though AutoMTL achieves better results, the number of model parameters is larger than the Multi-Task baseline. The inference time of AutoMTL in CityScapes is longer than Multi-Task and AdaShare.**
>
> **A1**: When designing multi-task models, **there is a tradeoff between the task accuracy, the model size, and the inference time**. The problem is fundamentally a multi-objective optimization problem. **AutoMTL aims to optimize for both task accuracy and model size, instead of only a single objective.**
>
> Specifically, in the multi-task baseline, all the parameters in the backbone are shared across tasks. Thus, the multi-task baseline always achieves the minimum in inference time and model size compared to independent models. None of the other baselines can compete with the multi-task baseline in terms of inference time and model size.
>
> However, the multi-task baseline usually suffers from significant task accuracy degradation due to task interference. **How to improve task accuracy while still offering resource efficiency is one of the fundamental motivations for AutoMTL.** AutoMTL demonstrates that, by allowing a small amount of task-specific parameters, we can significantly improve multi-task models’ accuracy. For example, compared to the multi-task baseline, AutoMTL increases averaged task performance by 10.1%, 7.0%, and 5.4% on CityScapes, NYUv2, and Taskonomy respectively, with an addition of only 18%, 21%, and 30% of model parameters (see Tables 1-3).
>
> AdaShare has the same optimization objective as AutoMTL – that is, optimizing both task accuracy and model size. AdaShare packs all tasks into a single backbone model and uses skip connections to mitigate task interference. Because of the skip connections, AdaShare has some inference latency improvement (e.g., 56.79ms on CityScapes) compared to independent models (71.01ms). AutoMTL addresses task interference by designing a more flexible search space. Compared to AdaShare, the different search space in AutoMTL increases average task performance by 2.3% on CityScapes, with a slightly increased inference latency (60.84ms).
>
> We want to emphasize that **the tradeoff between model size, task accuracy, and inference latency can be controlled by the regularization hyper-parameter $\lambda_{reg}$ in AutoMTL.** Using a larger $\lambda_{reg}$ can encourage more parameter sharing across tasks, further decreasing the model size and inference latency.  The following table shows more quantitative results on CityScapes with different $\lambda_{reg}$ to show the flexibility of AutoMTL. We also list the Single-Task model and AdaShare as a comparison.
>
> |  Model  | # Params (M) | Inference Time (ms) | Seg. mIoU $\uparrow$ |  Seg.  Pixel Acc. $\uparrow$ |    $\Delta t_1 \uparrow$                   | Depth Abs. Error  $\downarrow$ |  Depth Rel. Error  $\downarrow$    | Depth  $\delta < 1.25 \uparrow$ |    Depth $\delta < 1.25^2 \uparrow$    |   Depth  $\delta < 1.25^3 \uparrow$    |   $\Delta t_2 \uparrow$ | $\Delta t \uparrow$ |
> |:---------------------:|:------------:|:---------------------:|:----------:|:---------------------:|:----------------:|:----:|:-------:|:------:|:------:|:---------------------:|:------------------:|:------------:|
> |Single-Task  |   42.569   |   71.01   |   36.5   |   73.8   |     -     |   0.026   |   0.38   |   57.5   |   76.9   |   87.0   |     -     |     -     |
> |   AdaShare  | **21.285** |   56.79   |   40.6   |   74.7   |    +6.2   | **0.018** |   0.37   | **71.4** |   86.8   |   93.1   |   +15.5   |   +10.9   |
> |  w/ 0.01  |   23.626   | **56.38** | **43.4** | **74.9** | **+10.2** |   0.021   |   0.36   |   68.4   |   85.5   |   92.7   |   +12.2   |   +11.2   |
> |  w/ 0.001  |   25.584   |   61.08   |   43.3   |   74.8   |   +10.0   |   0.020   |   0.34   |   71.1   | **87.5** | **93.7** |   +15.7   |   +12.9   |
> | w/ 0.0005 (reported in the paper) |   28.819   |   60.84   |   42.8   |   74.8   |    +9.3   | **0.018** | **0.33** |   70.0   |   86.6   |   93.4   | **+17.1** | **+13.2** |
> |   w/ 0.0001  |   30.735   |   62.52   |   40.4   |   74.4   |    +5.7   |   0.019   |   0.37   |   68.1   |   84.5   |    92    |   +12.7   |    +9.2   |
>
> **Q2: How long does it take to train the model?**
>
> **A2**: The time cost of each training stage in terms of GPU hours is reported in the following table. The experiments were conducted on an Nvidia RTX8000.
>
> | Stage | CityScapes | NYUv2 | Taskonomy |
> |:------------:|:----------:|:-----:|:---------:|
> |   pre-train  |  8-9 | 20-21 | ~25 |
> | policy-train |12-13 | 36-37 | ~120 |
> |  post-train  | 14-15 | 44-45 | ~140 |
>
> And the time cost compared with AdaShare is shown in the table below. **AutoMTL doesn’t require a significantly higher cost even though our search space is much larger.**
>
> | Model | CityScapes | NYUv2 | Taskonomy |
> |:--------:|:----------:|:-----:|:---------:|
> | AdaShare | ~30|  ~70 | ~222 |
> |  AutoMTL | ~37 | ~103 |  ~285 |

---

> > ### Comment · Reviewer_koYc · 2022-08-09
> > **Thank you for the response**
> >
> > Thank you for your response. After reading your experiments and analysis, I have no more questions. I will keep the score unchanged.

---

> ### Author Response · Authors · 2022-08-08
> **Look Forward to Discussing with You**
>
> Dear reviewer koYc:
>
> We sincerely thank you for the review and comments. We have provided corresponding responses and results, which we believe have covered your concerns. We hope to further discuss with you whether or not your concerns have been addressed. Please let us know if you still have any unclear parts of our work.
>
> Best,
> Authors of Paper 6905

---

> ### Author Response · Authors · 2022-08-08
> **The Last 24 hours for Discussion**
>
> Dear reviewer koYc:
>
> We have provided corresponding responses and results, which we believe have covered your concerns. Notice that there are less than 24 hours left for the discussion period. We really hope to confirm with you whether or not your concerns have been addressed. Thank you very much!
>
> Best,
> Authors of Paper 6905

---

### Official Review · Reviewer_av4d · 2022-07-11

**Rating:** 6
**Confidence:** 4
**Soundness:** 4 excellent
**Presentation:** 3 good
**Contribution:** 3 good

**Summary:**

The authors propose a framework for automatically designing and training multi-task models. Given a set of tasks and a particular backbone network, the framework learns when to share layers for the best performance. For each task, the option is given to either use a shared version of a backbone operation, use a task-specific version, or skip the operation altogether. The network is trained with Gumbel-Softmax to learn in a differentiable way which of the three choices to make for every operation. At the end, discrete decisions are assigned and the final network design is trained from scratch. This approach performs well against other multi-task methods on the benchmarks CityScapes, NYUv2, and Taskonomy.


**Questions:**

- Not sure a definition of "operator" is ever provided? Is each parameterized operation (conv layer, normalization function) treated separately?
- Might be nice to see absolute numbers for # params in addition to relative percent drop in Tables 1-3

**Limitations:**

limitation + social impact discussion is included in the paper

**Strengths And Weaknesses:**

Strengths:

- The authors propose a straightforward and well-motivated search space for multi-task architectures, and show how to produce assignments in this space for an architecture that performs well against other multi-task methods while not requiring much additional parameter overhead.
- The approach can easily be adopted with new models, and the authors seem to have put a lot of emphasis on coding up a framework that’s easy to use

Weaknesses:

There are a couple other ablations that would have been nice to see to round out the experiments and get a clearer impression of the benefits of the proposed approach:

- I’m curious about the performance of a baseline like [i] trained with Gumbel-Softmax instead of RL. In contrast to this paper with 1 shared op + independent task-specific ops, the routing can instead be learned through some number of available shared ops.
- Is the three-stage training process necessary? How important is the initialization with the pre-trained model before beginning to learn the policy? Does the final model need to be trained from scratch or can the discrete subset of layers be sampled from the “policy-train” stage and finetuned from there?
- The regularization to encourage parameter sharing makes sense and seems to help from the ablations, I wonder about the weighting to encourage sharing particularly in the initial layers, how much does that help?

I don’t necessarily expect that all of those experiments would have been done or need to be done, just that the paper would be stronger with a few extra insights into the final design.

[i] Rosenbaum, Clemens, Tim Klinger, and Matthew Riemer. "Routing networks: Adaptive selection of non-linear functions for multi-task learning." ICLR 2018

Overall:

The proposed method is a nice contribution in the space of multi-task architecture search, the approach itself is not some huge, novel step away from existing methods, but it is well thought out and well executed with a clear paper and solid results.

---

> ### Author Response · Authors · 2022-08-02
> **Responses to Reviewer av4d**
>
> Thank you for your positive comments and valuable suggestions!
>
> **Q1: The performance of a baseline like [i] trained with Gumbel-Softmax instead of RL.**
>
> **A1**: We agree with you that it is feasible to replace the RL part in [i] with the differentiable neural architecture search with Gumbel-Softmax. And as you pointed out, their search space, which provides several available shared operators, differs from ours. **It would be a good future extension to integrate their search space into our framework**, leading to T operators plus skip-connections for each task to choose from. Whether the operators are shared or not depends on the learned policy. Thank you for your suggestion!
>
> **Q2: Is the three-stage training process necessary?**
>
> **A2**: Following your advice, we conduct an additional ablation study on the training process. Specifically, the necessity of the policy-train stage is verified in Section 4.2 already, so we focus on the pre-train and the post-train stages this time. We have added the details in Appendix Section E. Below is a summary.
>
> The quantitative results on CityScapes with and without the pre-train stage are shown in the table below. The results are collected using the same hyper-parameter setting reported in the paper. We can see that **AutoMTL with pre-training can obtain higher task performance than the model without pre-training**. This observation echoes existing work in NAS [1], which also suggests warming up the supermodel first and then conducting searching. Both ablation studies in [1] and our empirical study demonstrate that such a pre-train stage produces a better initialization for the parameters of the supermodel and eventually results in a more accurate multi-task architecture with a similar amount of parameters.
>
> |         Model         | # Params (M) | Seg. mIoU $\uparrow$ |  Seg.  Pixel Acc. $\uparrow$        |    $\Delta t_1 \uparrow$                   | Depth Abs. Error  $\downarrow$ |  Depth Rel. Error  $\downarrow$    | Depth  $\delta < 1.25 \uparrow$       |    Depth $\delta < 1.25^2 \uparrow$    |   Depth  $\delta < 1.25^3 \uparrow$    |     $\Delta t_2 \uparrow$                   | $\Delta t \uparrow$ |
> |:---------------------:|:------------:|:---------------------:|:----------:|:---------------------:|:----------------:|:----:|:-------:|:------:|:------:|:---------------------:|:------------------:|
> |    Single-Task  |    42.569   |     36.5     |    73.8    |     -      |       0.026      | 0.38 |   57.5  |  76.9  |  87.0  |      -     |     -     |
> | AutoMTL w/o pre-train |    28.878|   41.1      |    74.5    |          +6.8         |       0.020      | 0.41 |   65.7  |  82.7  |  90.6  |          +8.2         |        +7.5        |
> |  AutoMTL w/ pre-train |    **28.819**    |      **42.8**    |    **74.8**    |          **+9.3**         |      **0.018**      | **0.33** |   **70.0**  |  **86.6**  |  **93.4**  |      **+17.1**    |   **+13.2**   |
>
> The post-train stage can either use fine-tuning or training-from-scratch. The following table compares the task performance of the two options on the identical sampled multi-task architecture under the same hyper-parameter setting. The results show that **re-training the identified multi-task model from scratch would produce higher task performance**, which suggests retraining as a better post-train strategy. The observation is consistent with that of the well-known differentiable NAS method DARTS [2], which has become a common practice in recent years [1, 3, 4]. Note that in our paper, we also retrain all baselines from scratch for a fair comparison.
>
> |         Model      |  Seg. mIoU $\uparrow$ |  Seg.  Pixel Acc. $\uparrow$        |    $\Delta t_1 \uparrow$       | Depth Abs. Error  $\downarrow$ |  Depth Rel. Error  $\downarrow$    | Depth  $\delta < 1.25 \uparrow$       |    Depth $\delta < 1.25^2 \uparrow$    |   Depth  $\delta < 1.25^3 \uparrow$    |     $\Delta t_2 \uparrow$           | $\Delta t \uparrow$ |
> |:---------------------:|:------------:|:---------------------:|:----------:|:---------------------:|:----------------:|:----:|:-------:|:------:|:------:|:---------------------:|
> |   Single-Task    |   36.5   |   73.8   |     -    |   0.026   |   0.38   |   57.5   |   76.9   |   87.0   |     -     |     -     |
> | post-train w/ fine-tune |   41.8   |   74.6   |   +7.8   |   0.019   |   0.38   |   68.1   |   84.8   |   92.2   |   +12.3   |   +10.1   |
> |  post-train w/ retrain  | **42.8** | **74.8** | **+9.3** | **0.018** | **0.35** | **69.7** | **85.7** | **92.9** | **+15.6** | **+12.5** |
>
> [1] X. Zhang, et al. You only search once: Single shot neural architecture search via direct sparse optimization. T-PAMI2020.
> [2] Liu H, et al. Darts: Differentiable architecture search. ICLR2019.
> [3] Wu B, et al. Fbnet: Hardware-aware efficient convnet design via differentiable neural architecture search. CVPR2019.
> [4] Zela A, et al. Understanding and robustifying differentiable architecture search. ICLR2020

---

> > ### Author Response · Authors · 2022-08-02
> > **Responses to Reviewer av4d**
> >
> > **Q3: How does the policy regularization of encouraging sharing, particularly in the initial layers, help?**
> >
> > **A3**: To illustrate the benefits of the proposed policy regularization, we provide more quantitative results on CityScapes with different $\lambda_{reg}$ in the following table. All the other experiment settings including the training pipeline and the hyper-parameter setting remain the same.
> >
> > |         Model         | # Abs. Params (M) | # Rel. Params (%) | Seg. mIoU $\uparrow$ |  Seg.  Pixel Acc. $\uparrow$        |    $\Delta t_1 \uparrow$                   | Depth Abs. Error  $\downarrow$ |  Depth Rel. Error  $\downarrow$    | Depth  $\delta < 1.25 \uparrow$       |    Depth $\delta < 1.25^2 \uparrow$    |   Depth  $\delta < 1.25^3 \uparrow$    |     $\Delta t_2 \uparrow$                   | $\Delta t \uparrow$ |
> > |:---------------------:|:------------:|:---------------------:|:----------:|:---------------------:|:----------------:|:----:|:-------:|:------:|:------:|:---------------------:|:------------------:|:------------:|
> > | Single-Task |   42.569   |     -     |   36.5   |   73.8   |     -     |   0.026   |   0.38   |   57.5   |   76.9   |   87.0   |     -     |     -     |
> > | AutoMTL  w/ $\lambda_{reg}=0.01$   | **23.626** | **-44.5** | **43.4** | **74.9** | **+10.2** |   0.021   |   0.36   |   68.4   |   85.5   |   92.7   |   +12.2   |   +11.2   |
> > |  AutoMTL  w/ $\lambda_{reg}=0.001$  |   25.584   |   -39.9   |   43.3   |   74.8   |   +10.0   |   0.020   |   0.34   | **71.1** | **87.5** | **93.7** |   +15.7   |   +12.9   |
> > | AutoMTL  w/ $\lambda_{reg}=0.0005$   |   28.819   |   -32.3   |   42.8   |   74.8   |    +9.3   | **0.018** | **0.33** |   70.0   |   86.6   |   93.4   | **+17.1** | **+13.2** |
> > |  AutoMTL w/ $\lambda_{reg}=0.0001$   |   30.735   |   -27.8   |   40.4   |   74.4   |    +5.7   |   0.019   |   0.37   |   68.1   |   84.5   |    92    |   +12.7   |    +9.2   |
> >
> > **As the $\lambda_{reg}$ becomes larger, the probability of parameter sharing, especially those in initial layers, is higher, leading to the fewer number of parameters in the identified multi-task model but relatively lower task performance because of task interference.**
> > Users could adjust $\lambda_{reg}$ to control the tradeoff between resource efficiency and task accuracy. If the computation budget is limited, they could use a larger $\lambda_{reg}$ for a more compact model while sacrificing task accuracy. If the users call for the best task performance, they could tune $\lambda_{reg}$ to find the optimal setting.
> >
> > **Q4: The definition of "operator".**
> >
> > **A4**: An operator refers to a computation operation (e.g., Conv, BN, ReLU, etc.) in the computational graph of the backbone model (described in Line 51). When compiling to the supermodel, each parameterized operator is treated separately.
> >
> > **Q5: Might be nice to see absolute numbers for # params in Tables 1-3.**
> >
> > **A5**: Thank you for your suggestion! We have revised our paper to include the absolute numbers of parameters in Tables 1, 6, and 7. Parameters in task-specific heads are excluded. The following tables report only the absolute # params (M) for reference. A full version of the table is in the revised paper.
> >
> > |   Dataset  | Single-Task | Multi-Task | Cross-Stitch |  Sluice | NDDR-CNN |  MTAN  |   DEN  | AdaShare | AutoMTL |
> > |:----------:|:-----------:|:----------:|:------------:|:-------:|:--------:|:------:|:------:|:--------:|:-------:|
> > | CityScapes |    42.569   |   21.285   |    42.569    |  42.569 |  44.059  | 51.296 | 23.838 |  21.285  |  28.819 |
> > |    NYUv2   |    63.855   |   21.285   |    63.855    |  63.855 |  67.047  | 66.217 | 23.838 |  21.285  |  35.056 |
> > |  Taskonomy |   106.424   |   21.285   |    106.424   | 106.424 |  115.151 | 95.994 | 23.838 |  21.285  |  53.106 |

---

> ### Author Response · Authors · 2022-08-08
> **Look Forward to Discussing with You**
>
> Dear reviewer av4d:
>
> We sincerely thank you for the review and comments. We have provided corresponding responses and results, which we believe have covered your concerns. We hope to further discuss with you whether or not your concerns have been addressed. Please let us know if you still have any unclear parts of our work.
>
> Best,
> Authors of Paper 6905

---

> ### Author Response · Authors · 2022-08-08
> **The Last 24 hours for Discussion**
>
> Dear reviewer av4d:
>
> We have provided corresponding responses and results, which we believe have covered your concerns. Notice that there are less than 24 hours left for the discussion period. We really hope to confirm with you whether or not your concerns have been addressed. Thank you very much!
>
> Best,
> Authors of Paper 6905

---

### Author Response · Authors · 2022-08-06
**Look Forward to Discussing with You**

We would like to thank all the reviewers for their insightful comments and constructive suggestions. We have provided our detailed response to each reviewer and have also uploaded a revised version of our submission. We're open to receive any further questions and feedback.

Thank you for your time and participation!

---

### Meta-Review · Area_Chair_gfx4 · 2022-08-26

**Recommendation:** Accept
**Confidence:** Certain

**Metareview:**

All reviewers are positive about the paper. The AC concurs.

**Award:**

No

---

### Decision · Program_Chairs · 2022-09-14

Accept